# The Influence of 3D Printing Core Construction (Binder Jetting) on the Amount of Generated Gases in the Environmental and Technological Aspect

**DOI:** 10.3390/ma16165507

**Published:** 2023-08-08

**Authors:** Artur Bobrowski, Faustyna Woźniak, Sylwia Żymankowska-Kumon, Karolina Kaczmarska, Beata Grabowska, Michał Dereń, Robert Żuchliński

**Affiliations:** 1Faculty of Foundry Engineering, AGH University of Krakow, Reymonta 23 Str., 30-059 Kraków, Poland; wozfau@agh.edu.pl (F.W.); szk@agh.edu.pl (S.Ż.-K.); karolina.kaczmarska@agh.edu.pl (K.K.); beata.grabowska@agh.edu.pl (B.G.);; 2Bydgoszcz Cast Iron Foundry, Zygmunta Augusta 11 Str., 85-082 Bydgoszcz, Poland; odlewnia@odlewnia.com.pl

**Keywords:** 3D printing, binder jetting, thermal decomposition, emission of gases, furfuryl resin, castings defect

## Abstract

This article presents the findings of a study focusing on the gas generation of 3D-printed cores fabricated using binder-jetting technology with furfuryl resin. The research aimed to compare gas emission levels, where the volume generated during the thermal degradation of the binder significantly impacts the propensity for gaseous defects in foundries. The study also investigated the influence of the binder type (conventional vs. 3D-printed dedicated binder) and core construction (shell core) on the quantity of gaseous products from the BTEX group formed during the pouring of liquid foundry metal into the cores. The results revealed that the emitted gas volume during the thermal decomposition of the organic binder depended on the core sand components and binder type. Cores produced using conventional methods emitted the least gases due to lower binder content. Increasing Kaltharz U404 resin to 1.5 parts by weight resulted in a 37% rise in gas volume and 27% higher benzene emission. Adopting shell cores reduced gas volume by over 20% (retaining sand with hardener) and 30% (removing sand with hardener), presenting an eco-friendly solution with reduced benzene emissions and core production costs. Shell cores facilitated the quicker removal of gaseous binder decomposition products, reducing the likelihood of casting defects. The disparity in benzene emissions between 3D-printed and vibratory-mixed solid cores is attributed to the sample preparation process, wherein 3D printing ensured greater uniformity.

## 1. Introduction

Additive manufacturing, also known as 3D printing, is an innovative technology that has received a lot of attention in recent years in almost every field of science and industry. It involves a set of manufacturing processes that build 3D layered objects using digital design data. AM provides a number of benefits, such as increased design flexibility, minimized material waste, and the ability to produce complex geometries—unlike conventional subtractive manufacturing methods. To gain a comprehensive understanding of AM technology and explore its potential, various researchers have conducted extensive reviews in the field. Notably, Smith et al. conducted a comprehensive investigation into architected materials for additive manufacturing, providing valuable insights into different materials and techniques used in AM, as well as showcasing its potential for innovative engineering applications and groundbreaking advancements [1]. Furthermore, AM’s application extends beyond mere prototyping, as evident from Ahmad et al.’s recent study. Their research delved into the mechanical, thermal, and physical characteristics of oil-palm-fiber-reinforced thermoplastic composites, specifically tailored for Fused Deposition Modeling (FDM) 3D printers, thus introducing opportunities for sustainable material utilization and eco-friendly additive manufacturing [2]. Additionally, exploring diverse approaches to AM is critical to advancing the technology further. A study conducted by Mostafaei et al. focused on binder-jet 3D printing, analyzing process parameters, materials, properties, modeling, and the challenges associated with this technique [3]. Similarly, Wang et al. contributed to the optimization of laser powder bed fusion in AM. Their work provided an overview of melt pool characteristics, encompassing single- and multi-track melt pools, to support the process optimization of this technology [4].

In the field of 3D printing, molding and core sands containing furfuryl resins have also become a popular choice for casting production. Their widespread use is due to their favorable mechanical and technological properties, as well as their ability to accurately reproduce complex geometric shapes. As a result, these sands have found significant use in additive manufacturing (AM) processes for making molds and cores [5,6,7]. The foundry engineering interest in three-dimensional printing molds and cores for metal casting foundries is gaining popularity due to the increasing precision of printing. This is particularly due to the almost unlimited possibilities of reproducing intricate shapes, which would be very difficult or even impossible to achieve using conventional technology. Furthermore, 3D printing allows for much smaller dimensional tolerances, resulting in greater accuracy of the final product. One of the most significant advantages is also the speed of production, particularly for prototyping or small-scale casting. The printing process is fully automated, enabling the complete elimination or reduction in the number of cores, or simultaneous printing with the remaining part of the mold, which significantly shortens the production time compared to conventional production of foundry cores using core shooters, where each core is produced separately. In the case of complex castings, several or even dozens of cores may be required, which need to be joined (bonded) and placed in the mold cavity, significantly extending the production time [8,9]. An additional advantage of 3D-printed cores and foundry molds using furfuryl resin is their mechanical and technological properties. As studies have shown [8,10,11], 3D-printed cores using furfuryl resin exhibit the highest mechanical strength compared to cores made through conventional technology such as vibratory compaction, with the same binder content in the core sand. This results in low wear resistance and resistance to thermal deformation. Three-dimensional-printed cores also demonstrate resistance to mechanical damage caused by the erosive action of liquid foundry metal and metallostatic pressure, ensuring greater dimensional accuracy and structural stability of the castings, especially under high-temperature conditions. Sivarupan et al. [6] investigated the effect of process parameters on the flexure strength and gas permeability of 3D-printed sand molds, which are crucial for enhancing mold performance and material selection. Upadhyay et al. [7] presented a comprehensive review of 3D printing for rapid sand casting, summarizing the advancements, challenges, and potential applications in the manufacturing industry. Mitra et al. [9] focused on the rapid manufacturing process of functional 3D-printed sand molds, highlighting the importance of functional properties and their relevance to specific industrial needs. Finally, Xie et al. [8] conducted research on the influence of the addition of furan resin on the performance and accuracy of 3D-printed sand molds, contributing to the understanding of material choices and their impact on mold quality. Another crucial technological aspect involves developing a porous structure within 3D-printed molds and cores. This feature facilitates effective heating and cooling of these molds and cores. The 3D printing process, which involves applying successive layers of resin onto the previously coated binder grains, allows for the formation of micro-pores that facilitate the free flow of heat and achieve optimal casting properties by reducing shrinkage and minimizing thermal stresses during solidification [12,13,14].

An important technological and environmental issue related to 3D printing and application, especially in the context of generated gases, includes several key aspects that require attention and actions to minimize their negative impact on the environment [15,16]:Emission of toxic gases: During 3D printing using furfuryl resins, the thermal degradation of cores and molds results in the emission of harmful gases. These gas by-products may contain toxic organic compounds such as BTEX (benzene, toluene, ethylbenzene, and xylenes), which are hazardous to human health and the natural environment.Potential negative effects on ecosystems: If gas emissions are not adequately controlled, they can enter the natural environment, affecting ecosystems and their functions. Particularly in 3D printing for metal casting, where synthetic resins are used, potential pollutants can harm plants, animals, and soil.Air tightness and air pollution: Gas emissions during 3D printing can affect the air quality in the workplace and the surrounding area of the manufacturing facility. Exposing workers to harmful substances can lead to health issues, and improperly vented or polluted gases can negatively impact the air quality in the vicinity.Health implications for workers: Gas emissions during 3D printing can pose a health risk to workers involved in the manufacturing process. Harmful substances such as toxic fumes or dust can cause respiratory problems, skin and eye irritations, and even contribute to the development of more severe diseases.Improper waste management: Three-dimensional printing generates waste in the form of used resins and materials. Improper waste management can lead to increased amounts of waste containing environmentally harmful substances, negatively affecting soil and groundwater.

In addition, a significant amount of gases is emitted from the cores or mold during liquid metal infusion, which can cause voids, pores, or other undesirable defects in the finished casting. The tendency to form gas defects is an important aspect that affects the quality and durability of the castings produced. Therefore, controlling and minimizing gas emissions during thermal degradation of the binder is extremely important to ensure high-quality castings and eliminate potential manufacturing defects [17].

The main efforts focus on modifying resin compositions, searching for new, more environmentally friendly curing agents, and optimizing the composition of molding and core sands to minimize the necessary amount of introduced binder while ensuring the required mechanical and technological properties of the molding sands. The provided citations cover various aspects of 3D printing technology for sand molds and their applications. In one study [18], the mechanical characterization of an anisotropic silica sand/furan resin compound induced by binder-jet 3D additive manufacturing technology was explored. Another study [19] investigated the thermal properties of 3D-printed sand molds. Sivarupan et al. [20] presented research on reducing material consumption and hazardous chemicals through 3D printing of sand molds for energy-efficient metal part production. Hackney and Wooldridge [21] conducted a characterization of the direct 3D sand printing process for producing sand cast mold tools. Kim et al. [22] focused on fabricating a ceramic core for an impeller blade using a 3D printing technique and inorganic binder. Other studies involve smart molds for sand casting [23], multidisciplinary reviews on zero-waste manufacturing [24], experimental and numerical characterization of sand molds produced by additive manufacturing [25], and the preparation method of furan resin for 3D printing [26]. New proposals [27] have emerged to address the gas generation issue in printed molding and core sands by using mineral salts, which can help neutralize gases. Some of these salts can act as catalysts, accelerating the curing reaction and reducing the emission time of gas products from the molds. Another solution [15,28] is conducting core production under vacuum conditions, which can help reduce the amount of gases emitted during the curing process and ultimately during the interaction with the high temperature of the liquid foundry metal. The emissions of materials used in casting production are gaining importance, as reflected in publications dedicated to monitoring the composition of gas products resulting from the thermal degradation of molding and core sands [29,30,31,32]. This stems from the need to reduce the negative impact on the natural environment, as well as the working environment [33,34,35,36,37], and it is achievable through the use of appropriate resin quality, selecting the proper 3D printing temperature, or even choosing the 3D printer model [38]. Three-dimensional printing has enormous eco-friendly potential as it allows for the replacement of many costly and time-consuming production processes while also minimizing the amount of materials consumed, thereby reducing energy consumption and waste generation [39].

Despite the significant advantages of 3D printing of molds and cores from furfuryl resin in metal foundries, there is a lack of research on reducing the negative impact of this technology on the environment and volatile emissions. In particular, in-depth research is needed to understand and minimize the emission of harmful gases, especially toxic organic compounds such as BTEX (benzene, toluene, ethylbenzene, and xylenes), generated during thermal degradation. While the article briefly mentions some proposed solutions, such as modifying the resin composition, using mineral salts as catalysts, and producing cores under vacuum conditions, there is a need for more research and experimental data to confirm the effectiveness and practicality of these approaches in reducing gas emissions during 3D printing of molds and cores. Addressing this research gap is critical to promoting the sustainable and environmentally friendly potential of 3D printing in metal foundries, ensuring the safety and well-being of workers, and minimizing the environmental impact of the manufacturing process. Furthermore, through research and the implementation of effective solutions, we can significantly minimize the negative impact of the production process on the environment. This, in turn, is crucial for supporting sustainable practices in the foundry industry, ultimately contributing to a more ecological and responsible approach to manufacturing.

The study presents research outcomes comparing gas emissions during the thermal degradation of 3D-printed cores in binder-jetting technology with conventional cores made using commercial furfuryl resin, considering the type and amount of binder utilized. The analysis focused on the volume of generated gases and the quantification of BTEX compounds, which determine the environmental harmfulness of foundry sands and cores. Additionally, the research aimed to show the potential reduction in gas emissions from 3D-printed cores by employing shell cores with fewer organic additives responsible for gas generation. Minimizing gas production during the binder’s thermal decomposition will positively impact casting structure, homogeneity, durability, and mechanical properties. Notably, significant reduction in BTEX gas emissions during mold and core production will prevent the formation of defects such as voids and pores caused by inappropriate thermal degradation parameters. Consequently, reducing the release of volatile compounds becomes a crucial factor in improving casting quality within the foundry industry.

## 2. Materials and Methods

### 2.1. Materials

Two types of binders based on furfuryl resin were used for the research:Kaltharz U404—commercial resin hardened with an acid hardener 100T3 (manufacturer: Huttenes-Albertus, Düsseldorf, Germany); typical for conventional cores sand production;FB001—commercial resin hardened with an acid hardener FA001 (manufacturer: ExOne; Irwin, PA, US); suitable for making cores and molds in a wide variety of silica sand and ceramic material by binder-jetting 3D printing.

Table 1 summarizes the compositions of the core sands used in the study.

The selection of furfuryl binders for core sands research is driven by their unique and advantageous properties compared to other resin families. Furfuryl resins exhibit low viscosity, facilitating smooth handling and precise processing during 3D printing. Their high crosslinking efficiency enables rapid solidification, reducing the risk of distortion in printed cores. Additionally, furfuryl resins demonstrate good thermal stability, making them suitable for applications involving elevated temperatures. Furfuryl resins also possess inherent chemical resistance, rendering them suitable for contact with certain chemicals or corrosive environments. Lastly, their cost effectiveness compared to alternative resin materials in 3D printing makes them an attractive option for core sands research. By considering these properties, furfuryl binders offer immense potential for advancing core sands in 3D printing applications, justifying their selection for focused investigation.

### 2.2. Core Samples

The tests were conducted using standard cylindrical samples (Ø50 × 50 mm) produced by a conventional method using compaction and using 3D printing technology by ExOne. The method of obtaining cylindrical samples is described in [11]. Taking into account the variation in the shape structure, two types of cylindrical samples were prepared:Solid cores (Figure 1a)—obtained by conventional compaction or 3D printing;Shell cores filled with uncured sand (Figure 1b)—obtained only by 3D printing.

Figure 1 illustrates the appearance of the cylindrical samples used in the study.

Table 2 lists the types of shapes, taking into account the type of molding sand and technology of their manufacture.

Sand with hardener was removed from the MS-4.3 cores, and its place was filled with quartz sand fired at 1300 °C to fill the free air space, which could affect the accuracy of the measurements. The weight of the calcined quartz sand was not taken into account in the emission calculations.

### 2.3. Methodology

The procedure for evaluating the gas emission properties of molding compounds in the foundry industry plays a crucial role in assessing the integrity of cores and molds utilized in casting production. Gas emission testing aims to identify the quantity and composition of gases released during the heating or melting of molding sands, which can significantly impact the casting quality. To conduct these tests, a combination of experimental setups and gas chromatography is employed, allowing for the quantification and separation of different gaseous components from molding sand samples. This enables the monitoring of potential emissions of environmentally or occupationally hazardous gases, facilitating adjustments in molding sand formulations to minimize their gas-forming characteristics. The research methodology is illustrated schematically in Figure 2.

The following subsections discuss the research process in more detail.

#### 2.3.1. The Gas Emission Determination

The research on the volume and kinetics of gas emission was conducted on a device developed at the Foundry Department of AGH University of Science and Technology and patented (patent no. PL 224705, B1), the schematic of which is shown in Figure 3.

The sample of the tested core sand, in the form of a cylinder with dimensions Ø50×50 mm, previously weighed, is poured with liquid casting alloy (cast iron) at a temperature of 1350 °C. The mass of the liquid metal in the mold is approximately 9 kg. The mold, in which the tested sample mounted in a steel bell is placed, is made of molding sand with bentonite (mold weight approximately 24 kg). Gases produced as a result of the thermal degradation of the binder, which separate from the sample, pass from the steel bell to the capsule (Figure 4), where a gas drying system is located. Then, they are directed to an activated carbon bed where they undergo adsorption. The activated carbon bed (previously activated) is divided into two layers: safety/control (Figure 4; Activated carbon A) and measurement (Figure 4; Activated carbon B).

After passing through the capsule, gases are guided through a system of pipes to a peristaltic pump, which is properly calibrated and connected to a recorder that records the volume of emitted gases over time.

#### 2.3.2. Gas Chromatography

To carry out qualitative and quantitative studies of gases by gas chromatography, activated carbon from a capsule on which gases from the thermal decomposition of resin were adsorbed was used. The measurement layer (Figure 4; Activated carbon B) contains 700 mg of activated carbon and serves as the primary site for adsorption. The second layer (Figure 4; Activated carbon A) contains 200 mg of activated carbon and serves as a control, providing information about any breakthrough in layer B that may occur with cores exhibiting very high gas emissions. The layer of activated carbon with adsorbed organic chemicals was extracted with diethyl ether. The analysis of substances from the BTEX group was conducted using gas chromatography with a flame ionization detector (FID, Thermo Scientific, Waltham, MA, USA). The identification of BTEX was performed using a system consisting of a Trace GC Ultra gas chromatograph (Thermo Scientific) equipped with a capillary chromatographic column (Rtx^®^-5MS Columns; fused silica; low-polarity phase; crossbond diphenyl dimethyl polysiloxane; Restek™, Bellefonte, PA, USA) with a length of 30 m and an inner diameter of 0.25 mm [40].

## 3. Results

Table 3 presents the results of gas volume measurements during the thermal degradation of the binder, calculated per 1 kg of core sand, as well as the quantity of compounds from the BTEX group, for cores produced using conventional molding sand technology with furfuryl resin (vibration compaction). The standard binder addition consists of 1 part by weight of U404 resin and 0.5 parts by weight of 100T3 hardener (MS-1). Additionally, the table includes results for a core made with the same binder but with the same component ratio (resin, hardener) as in the case of 3D printing, namely, 1.5 parts of resin and 0.4 parts by weight of hardener.

The results indicate that both the quantity of organic components and the type of binder have an impact on the volume of gases produced during the thermal decomposition of the binder. The use of a larger amount of binder (sample MS-2) dedicated to conventional molding and core technology with furfuryl resin leads to an increase in gas volume of approximately 37% compared to the initial composition (sample MS-1). A similar trend was observed for benzene and toluene, with their content in the emitted gases increasing by approximately 27% and 41%, respectively.

Table 4 presents the results of gas volume measurements and the content of compounds from the BTEX group depending on the core design. The cores were made using 3D printing technology. The MS-4.1 core was a solid core, while the MS-4.2 and MS-4.3 cores were shell cores (according to Figure 1). However, it should be noted that the MS-4.2 core contained fused quartz sand FS-001 mixed with hardener FA-001 inside the core, while in the MS-4.3 core, the sand with hardener was removed and replaced with pre-heated quartz sand to fill the free space, which could affect the measurement accuracy.

From Table 4, it can be inferred that the use of shell cores allows for a reduction in gas emissions of over 20%, in the case of retaining the sand with hardener inside the core, and over 30% when using cores in which the sand with organic additives is removed (MS-4.3). The differences are significant enough to have a decisive impact on the propensity for gas-related defects in castings. In terms of the harmfulness of core sand, as indicated by the emission of compounds from the BTEX group, a 25% reduction in the emission of carcinogenic benzene was demonstrated for the MS-4.2 core and approximately 32% lower emission for the MS-4.3 core sand. From the perspective of increasing requirements related to atmospheric emissions, the obtained research results can be seen as a new trend in shaping core designs. However, it is important to ensure the preservation of appropriate technological and mechanical properties of the cores. Research conducted by the authors [11] showed that 3D-printed shell cores have advantageous parameters that meet the requirements of foundry plants. In addition, the use of shell cores allows the consumption of resin to be reduced, which is extremely beneficial from the point of view of the economics of the core production process, as the purchase price of a binder dedicated to 3D printing is high.

Figure 5, Figure 6 and Figure 7 show the average of three measurements of the dependence of the volume of emitted gases over time and the rate of their emission.

Reducing the overall gas generation during the 3D-printed core casting process by 20% to even 30% compared to solid cores can significantly impact defect reduction in castings. Gas porosities, bubbles, and punctures are common and challenging defects to eliminate. This is due to the necessity of ensuring sufficient mechanical strength of the core, which is exposed to high pressure from the liquid casting alloy.

Cores made from a reduced binder content core mixture will emit fewer gases, but often fail to meet the required strength. Therefore, a higher amount of organic additives is introduced, resulting in increased gas emissions. The use of shell cores is considered by the authors as one of the methods that can be used to limit gas formation, especially when produced using additive technologies.

If the shell core with a specified wall thickness possesses sufficient strength to withstand metallostatic pressure, it reduces both the binder amount and the tendency to create defects in castings. The presence of loosely packed quartz sand inside the shell core allows for easier gas evacuation compared to a solid core. By appropriately designing the core structure, there is a possibility to direct gases outside the mold.

The rate of emission of gaseous products resulting from thermal decomposition of the binder is about 0.26 dm^3^/g·s for the solid core (MS-4.1) and about 0.19 dm^3^/g·s for the shell core (MS-4.2). The highest rate of gas emission was recorded for the shell core filled with roasted quartz sand (MS-4.3)—0.29 dm^3^/g·s. It was also noticed that the thermal decomposition of the binder in the solid core (MS-4.1) lasts the longest (Figure 5a). Gaseous thermal degradation products are released about 800 s from the moment of first contact with the high temperature of the liquid casting alloy. In the case of the shell core (MS-4.2), the volume of released gases (Figure 6a) stabilizes after about 600 s, and the rate of gas emission over time is less rapid compared to that of the solid core (Figure 6b). In turn, for the shell core filled with roasted quartz sand (MS-4.3), the highest dynamics of the gas emission process was recorded, probably caused by the facilitated evacuation of gases from the core due to free spaces in the unbound quartz sand. The maximum volume of emitted gases (Figure 7a) is obtained after about 500 s, and the speed of emitted gases reaches a small value after about 150 s (Figure 7b), in contrast to the solid core, where this time is about 220–250 s.

The highest rate of gas release (outlet) for a shell core filled with roasted quartz sand may result from shock overheating of almost the entire thickness of the core with simultaneous facilitated flow of gaseous products through the inner part of the core (loosely buried sand). In the case of a solid core, the emission rate is only slightly slower, but the core is gradually heated to greater and greater depth, which favors the generation of more gases over a longer time interval. The heating effect was well captured during the registration in the form of a second intense peak 25 s after pouring the liquid casting alloy into the core. After 100 s, the rate is still about 0.1 dm^3^/gs. On the other hand, for a shell core filled with quartz sand, the emission rate is lower by half. An extended time of gas release from the core may have negative effects, as it may be trapped in the solidifying layer of castings and will not be able to escape due to the increasing density of the alloy over time. Taking into account the facilitated flow of gases through the interior of the core, the lower rate of evolution and the smaller volume of gases will help to reduce defects of gas origin.

## 4. Conclusions

Based on the conducted research, the following conclusions can be drawn:The volume of emitted gases generated during the thermal decomposition of organic binder (furfuryl resin, acid hardener), under the same conditions of pouring with liquid casting metal, depends on the quantity of components in the core sand and the type of binder.Cores made using conventional technology emit the least amount of gases due to the lowest binder content.Increasing the amount of Kaltharz U404 resin to 1.5 parts by weight (MS-2) results in a 37% increase in gas volume compared to the initial composition (MS-1) and an increase in benzene emission by 27%.The use of shell cores allows for a reduction in gas volume by over 20%, in the case of retaining sand with a hardener, and even by 30% in the case of cores where sand with a hardener is removed (MS-4.3).The use of shell cores is an environmentally friendly solution that enables a reduction in the emission of carcinogenic benzene by up to 30% and also allows for a reduction in core production costs due to decreased resin consumption.The use of shell cores promotes faster removal of gaseous binder decomposition products, and thus reduces the tendency to create defects in castings.

The authors plan to conduct tests to confirm the assumptions they have made. Model tests will be carried out on relatively simple cores, which can be prepared as shell cores, following the adopted methodology. A core with a shape that is too complicated will make it impossible to remove the sand with hardener from inside the core and to prepare a core filled with roasted quartz sand.

## Figures and Tables

**Figure 1 materials-16-05507-f001:**
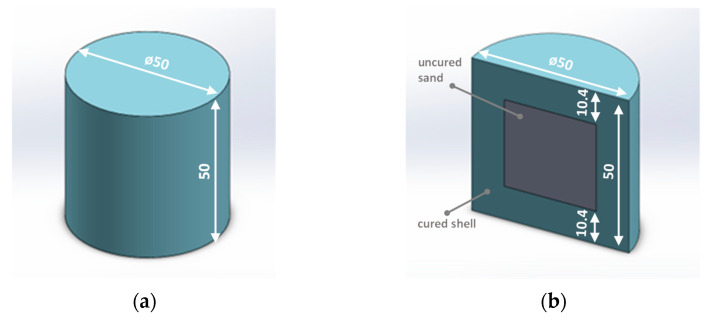
Cylindrical samples: (**a**) solid hardened throughout the sample volume; (**b**) shell filled with uncured sand with hardener (cross-sectional view; dimensions in mm) [11].

**Figure 2 materials-16-05507-f002:**
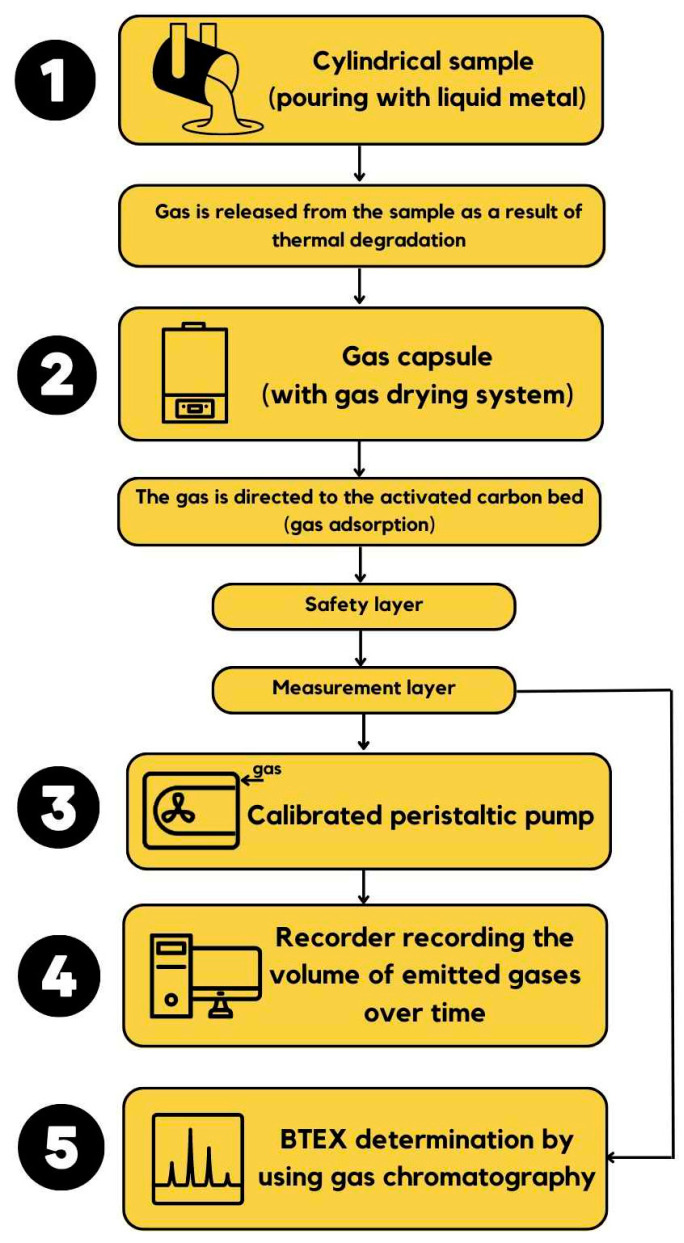
Scheme of the research procedure.

**Figure 3 materials-16-05507-f003:**
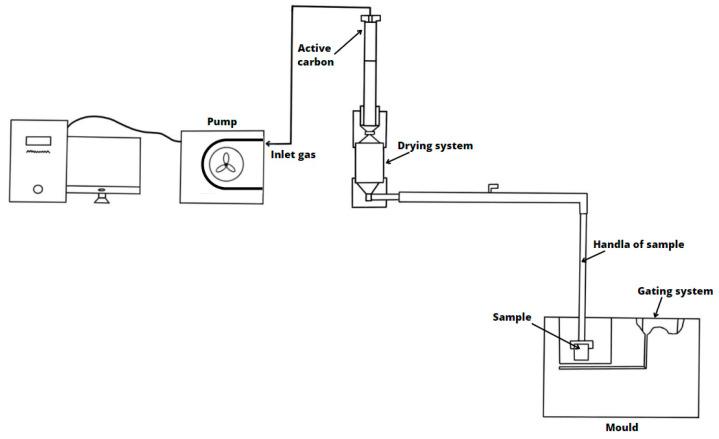
Experimental stand for the determination of the emitted gas volume and the BTEX emission. Based on [31].

**Figure 4 materials-16-05507-f004:**
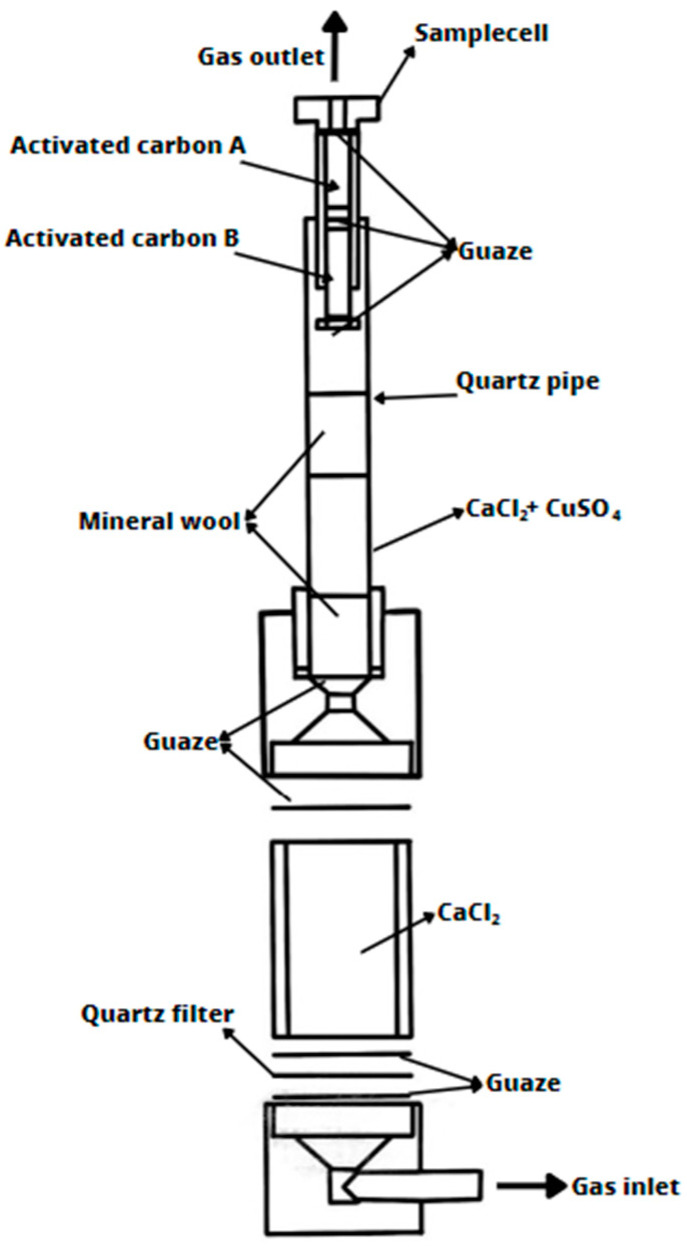
Capsule for sampling gases for the BTEX content. Based on [31].

**Figure 5 materials-16-05507-f005:**
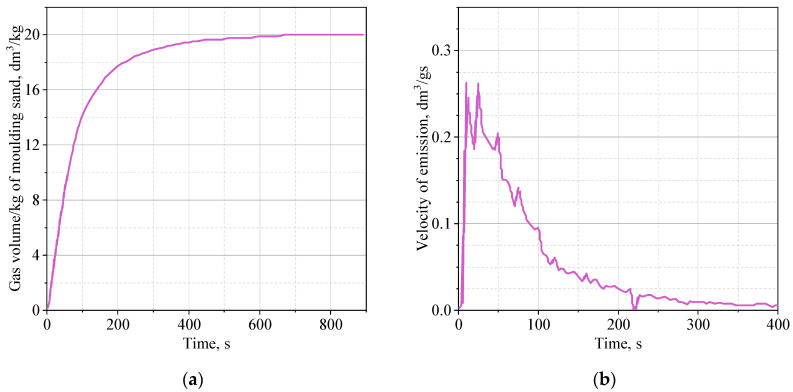
Data for MS-4.1 sample: (**a**) emission of gases during exposure to high temperature; (**b**) rate of emission of gases during exposure to high temperature.

**Figure 6 materials-16-05507-f006:**
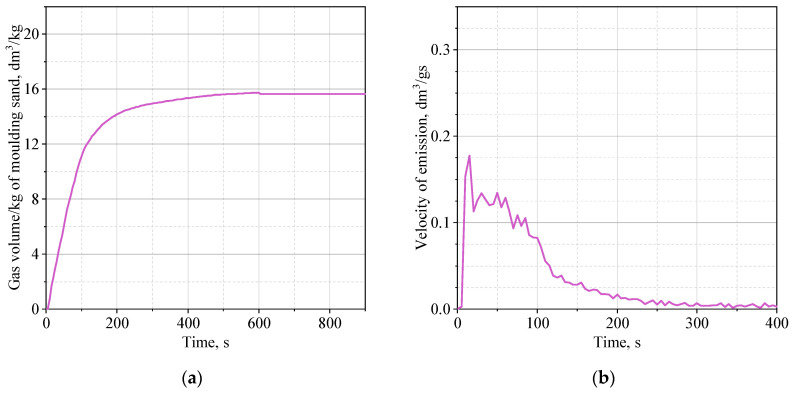
Data for MS-4.2 sample: (**a**) emission of gases during exposure to high temperature; (**b**) rate of emission of gases during exposure to high temperature.

**Figure 7 materials-16-05507-f007:**
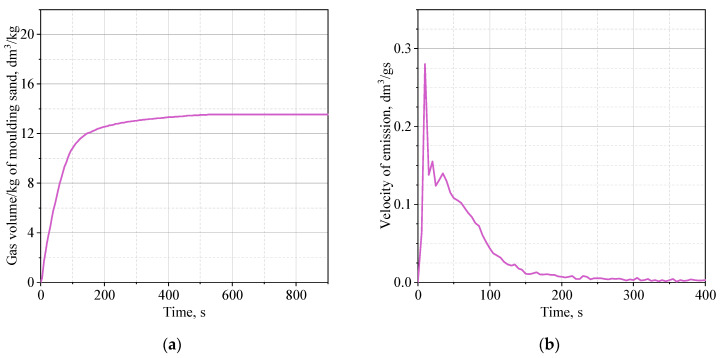
Data for MS-4.3 sample: (**a**) emission of gases during exposure to high temperature; (**b**) rate of emission of gases during exposure to high temperature.

**Table 1 materials-16-05507-t001:** Summary of molding sand systems tested for selected properties.

Molding Sands for Samples	A method of Samples Preparing	Binder ^a^, Part by Weight		Hardener, Part by Weight		Sand Grain Matrix, Part by Weight		
		FB001 ^b^	Kaltharz U404 ^c^	FA001 ^a^	Aktivator 100T3 ^b^	FS001 ^a^	Coarse Sand ^d^ (Main Fraction 0.40/0.32/0.20)	Sieved Sand ^d^ (Fraction 0.16/0.10)
MS-1	mixing /compacting	-	1.0	-	0.5	-	100	-
MS-2	mixing /compacting	-	1.5	-	0.4	-	-	100
MS-3	mixing /compacting	1.5	-	0.4	-	-	100	-
MS-4	3D printing (binder jetting)	1.5	-	0.4	-	100	-	-

^a^ Chemically cured furfuryl resins along with their acidic activators are commercially available products used in a variety of applications, including 3D printing; since both resins are commercial products, the manufacturer does not disclose the exact composition. ^b^ ExOne; ^c^ Huttenes-Albertus; ^d^ Sibelco Poland sp. z o.o., Bukowno, Poland.

**Table 2 materials-16-05507-t002:** Characteristics of the types of cores samples.

Sample	Type of Sample	Type of Sample Filling	Total Sample Weight ^d^, g	Mass of Roasted Quartz Sand in the Sample ^e^, g
MS-1	full sample	the same as the whole shape	153.7	-
MS-2	full sample	the same as the whole shape	138.2	-
MS-3	full sample	the same as the whole shape	138.0	-
MS-4.1	full sample	the same as the whole shape	131.7	-
MS-4.2	shell sample	quartz sand FS001 with hardener FA001	137.4	-
MS-4.3	shell sample	roasting quartz sand fraction 0.16/0.10	110.5	28.7

^d^ Sibelco Poland sp. z o.o. ^e^ Average of 3 samples.

**Table 3 materials-16-05507-t003:** Gas emission from molding sands MS-1–MS-3 (in mg/kg of molding sand), average of three measurements.

Sample Designation	Sample Weight, g	The Volume of Gases, dm^3^/kg Weight	Gas Emission, mg/Specimen			
			Benzene	Toluene	Ethylbenzene	Xylenes
MS-1	153.7	17.0	338.6	57.7	0.0	13.2
MS-2	138.2	23.4	431.0	81.8	0.0	12.4
MS-3	138.0	20.8	365.3	51.5	0.0	13.4

**Table 4 materials-16-05507-t004:** Gas emission from molding sand MS-4 taking into account the type of shape (in mg/kg of molding sand), average of three measurements.

Sample	Sample Weight, g	The Volume of Gases, dm^3^/kg Weight	Gas Emission, mg/Specimen			
			Benzene	Toluene	Ethylbenzene	Xylenes
MS-4.1	131.7	20.0	648.0	50.4	1.7	20.3
MS-4.2	137.4	15.7	488.1	32.4	0.8	20.1
MS-4.3	139.3	13.5	430.9	22.8	0.0	17.1

## Data Availability

The data is contained within the article and/or available on request from the corresponding author.

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
