# Peer review of "The Influence of 3D Printing Core Construction (Binder Jetting) on the Amount of Generated Gases in the Environmental and Technological Aspect"

_materials, 2023, doi:10.3390/ma16165507_

Round 1
Reviewer 1 Report
The authors studied the influence of 3D printing core construction (Binder Jetting) on the amount of generated gases in the environmental and technological aspect. The manuscript had an interesting topic and was well-written, however, it could only be accepted with the following corrections:
1. The abstract of the paper remains vague; where the results of the study were not highlighted clearly (empirical data is most preferred) and a brief summary was not found.
2. It is advised at the beginning of the Introduction Section the authors should define briefly about additive manufacturing (AM) technology and its advantages. Therefore, it is recommended the authors can add the following papers as references:
· Architected materials for additive manufacturing: A comprehensive review. Materials, 15(17), 5919. https://doi.org/10.3390/ma15175919
· Mechanical, thermal and physical characteristics of oil palm (Elaeis Guineensis) fiber reinforced thermoplastic composites for FDM–Type 3D printer. Polymer Testing, 120, 107972. https://doi.org/10.1016/j.polymertesting.2023.107972
3. In the introduction section, the research gap was not defined clearly.
4. Figure 1, please state the SI unit for the dimensions.
5. Refer to lines 133 and 134, is there any explanation for items 3 and 4?
6. For the experimental procedure, it is recommended the authors provide a flow chart for the experimental setup.
7. Refer to lines 199, 201, 207, and 208, it is better to state for one unit of time either second or minute.
8. Section 3 was missing to discuss or cite the previous studies (result comparison).
9. The conclusion could be improved by adding the future study, limitations and implications for researchers.
Author Response
Dear Reviewer,
thank you for your review, please find attached our responses.

Author Response

(The authors gave the same response as above.)

Reviewer 3 Report
The work presents a brief study on the effect of binder jetting on technology and environmental aspect. Although the authors have tried to write a good article but there are certain deficacy in the submitted manuscript and needs improvement as follows:
1. The abstract must contain some significant results from the study and not just texts. Numerical results would attract the research fraternity.
2. The introduction is poorly written. Too many refernces at one place (see line 49, 60, etc.)
3. the literature part in the introduction need serious modification. The research gap could not be stated properly.
4. the objective is not clear at all.
5. The actual composition of the resins must be presented in the study.
6. What are the characterization methods used in the study? Specification of equipment missing.
7. Fig 4-6 need in-depth discussion. The authors have barely discussed the graphs.
8. the conclusions must be concise and to the point. It looks lengthy and exaggerated.
There are numerous grammatical error throughout the manuscript. It needs serious revision prior to submission.
Author Response

(The authors gave the same response as above.)

Round 2
Reviewer 1 Report
A minor error was detected as below:
- Refer to lines 47 - 57; 'as evident from Johnson et al.' This cited reference was wrong if referring to ref no. [2] it is Ahmad et al. (please check).
Author Response
Dear Reviewer,
thank you for your kind commitment and detailed comments. The manuscript has been revised as suggested.
Reviewer 2 Report
Well done. The manuscript has been improved after the revision.
Author Response
Dear Reviewer,
thank you for your kind commitment and detailed comments.
Reviewer 3 Report
The authors have successfully addressed the comments, and the necessary changes have been incorporated in the manuscript as per the suggestions. The paper looks fine to me now.
Author Response

(The authors gave the same response as above.)
